# Towards Natural Image Matting in the Wild via Real-Scenario Prior

## Abstract

Recent approaches attempt to adapt powerful interactive segmentation models, such as SAM, to interactive matting and fine-tune the models based on synthetic matting datasets. However, models trained on synthetic data fail to generalize to complex and occlusion scenes. We address this challenge by proposing a new matting dataset based on the COCO dataset, namely COCO-Matting. Specifically, the construction of our COCO-Matting includes accessory fusion and mask-to-matte, which selects real-world complex images from COCO and converts semantic segmentation masks to matting labels. The built COCO-Matting comprises an extensive collection of 38,251 human instance-level alpha mattes in complex natural scenarios. Furthermore, existing SAM-based matting methods extract intermediate features and masks from a frozen SAM and only train a lightweight matting decoder by end-to-end matting losses, which do not fully exploit the potential of the pre-trained SAM. Thus, we propose SEMat which revamps the network architecture and training objectives. For network architecture, the proposed feature-aligned transformer learns to extract fine-grained edge and transparency features. The proposed matte-aligned decoder aims to segment matting-specific objects and convert coarse masks into high-precision mattes. For training objectives, the proposed regularization and trimap loss aim to retain the prior from the pre-trained model and push the matting logits extracted from the mask decoder to contain trimap-based semantic information. Extensive experiments across seven diverse datasets demonstrate the superior performance of our method, proving its efficacy in interactive natural image matting. Code is available in the supplementary file.

## 1 Introduction

Image matting (Ruzon & Tomasi, 2000; Levin et al., 2007) aims to predict the alpha mattes for foreground subjects, enabling their seamless extraction from complex backgrounds. This capability is indispensable for a multitude of applications, including film production, video editing, and graphic design (Li et al., 2023a). Mainstream matting methods (Yao et al., 2024a) take auxiliary trimaps as input, which divide the image into foreground, background, and unknown regions, and thus facilitate the image matting to the refinement of the unknown regions. Nonetheless, labeling trimaps is labour-consuming which limits their practical use. To address this, interactive matting (Ding et al., 2022) has emerged as a solution, replacing trimaps with more accessible cues such as bounding boxes (BBox), points, or scribbles. Among these, BBox are particularly advantageous because of their ease of acquisition, superior accuracy relative to other prompts (Ye et al., 2024), and compatibility with object detection networks for automatic matting on predefined classes (Li et al., 2024). Thus, our work primarily focuses on interactive matting using BBox as the auxiliary input.

Unfortunately, due to the hard-to-obtain alpha matte annotations, existing interactive matting methods (Li et al., 2024) combine an alpha matte with multiple background images (Ye et al., 2024) to generate synthetic training data. These synthetic data suffer from a distribution discrepancy compared to natural data, hindering the network's generalization to natural scenes. To overcome this, recent methods have sought the help from the robust pre-trained networks, like Segment Anything Model (SAM) (Kirillov et al., 2023) which is trained on one billion real-world masks. For instance, approaches like MatAny (Yao et al., 2024b) and MAM (Li et al., 2024) utilize SAM's intermediate features or segmentation masks to construct an additional matting network.

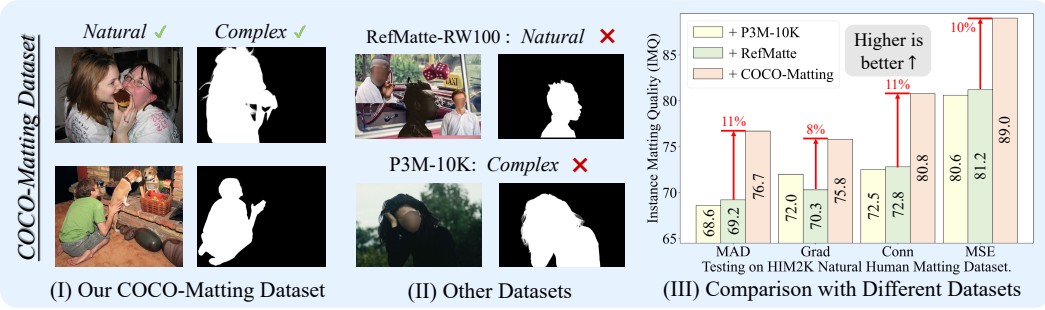

(a) Our COCO-Matting features both natural and complex scenes. Training SEMat with COCO-Matting enables a substantial lead on HIM2K than using RefMatte (Li et al., 2023b) and P3M-10K (Li et al., 2021a).

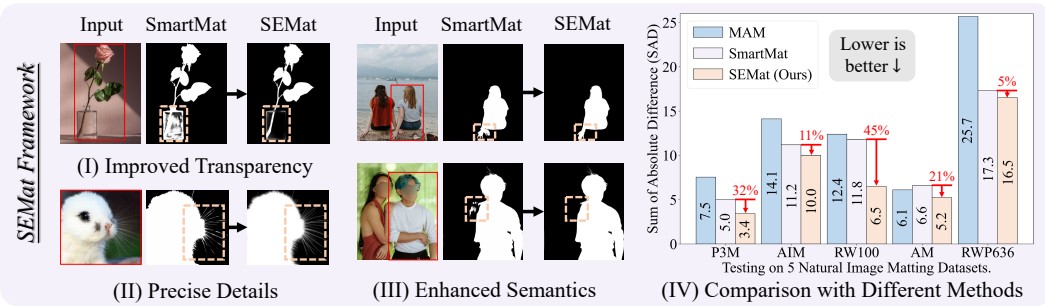

(b) Our SEMat trained on COCO-Matting significantly outperforms the SoTA methods like MAM (Li et al., 2024) and SmartMat (Ye et al., 2024) on several datasets, especially in transparency, details, and semantics.

Figure 1: Improvements of our proposed COCO-Matting dataset and SEMat framework.

However, the SAM-based approaches have not fully exploited the potential of pre-trained SAM, and are encumbered by several critical issues. **(1) Inappropriate Data**. Training on synthetic datasets like RefMatte (Li et al., 2023b) or simple natural datasets like P3M-10K (Li et al., 2021a) in Figure 1a is insufficient for robust generalization. This limitation not only impedes the model's adaptation to diverse natural scenes like HIM2K dataset (Sun et al., 2022a) but also risks undermining the diverse pre-trained knowledge inherent in SAM. **(2) Feature Mismatch**. The intermediate features and predicted masks from a frozen SAM are not optimally aligned with the nuanced demands of matting which necessitates an enhanced sensitivity to edge refinement and transparency perception, as well as the mask prediction for transparent matting-specific objects that are not common in segmentation datasets. **(3) Loss Constraints**. Traditional matting losses supervise the network to learn the synthetic training data, neglecting the broader imperative of generalization across natural images. This oversight limits the model's applicability and efficacy in real-world scenarios.

**Contribution.** To solve these challenges, we propose the COCO-Matting dataset consisting of complex natural images and Semantic Enhanced Matting (SEMat) framework to revamp training datasets, network architecture, and training objectives, greatly improving the interactive matting performance. Our main contributions are highlighted below.

Firstly, to solve the first challenge of inappropriate data, we construct the COCO-Matting dataset based on the renowned COCO dataset (Lin et al., 2014). Considering the complicated interactions between human and their surrounding environment, we focus on humans as the primary subjects and create 38,251 instance-level alpha mattes. Notably, as shown in Figure 1a, our dataset is unique in featuring complex natural scenarios, setting it apart from others. Specifically, the construction of our COCO-Matting is composed of Accessory Fusion and Mask-to-Matte. (1) Accessory Fusion aims to solve the problem of missing accessories for human mask annotations through the overlap rate between masks, i.e., merging the accessory masks, such as hats, backpacks, or items being held that are considered part of human in matting tasks. (2) For Mask-to-Matte, it is proposed to solve the problem of coarse mask annotations by converting binary masks into continuous high-precision alpha mattes with a trained trimap-based network (Hu et al., 2023) to match the matting annotations.

Secondly, to address the second and third challenges, we design a novel and effective SEMat framework which improves the network architecture and training objectives. (1) Our network in SEMat

consists of the Feature Aligned Transformer (FAT) and Matte Aligned Decoder (MAD). For FAT, it is to solve the problem of unaligned features between segmentation and matting through the introduced prompt enhancement and Low-Rank Adaptation (LoRA) (Hu et al., 2022) on the transformer backbone. Regarding MAD, it aims to predict aligned mattes on matting-specific objects such as smoke, nets, and silk by the learnable matting tokens, matting adapter, and lightweight matting decoder. (2) We design more effective training objectives, including the regularization loss and trimap loss. Among them, the regularization loss ensures the consistency of predictions between the frozen and learnable networks to retain the prior of the pre-trained model; the trimap loss aims to encourage the matting logits extracted from the mask decoder to contain trimap-based semantic information.

Finally, experiments show the significant improvement of our COCO-Matting dataset and SEMat framework compared with the state-of-the-arts (SoTAs). For example, Figure 1a shows that when combining the same 3 synthetic datasets Distinction-646 (Qiao et al., 2020), AM-2K (Li et al., 2022), and Composition-1k (Xu et al., 2017) with RefMatte (Li et al., 2023b) or P3M-10K (Li et al., 2021a) or our COCO-Matting to train our SEMat, our COCO-Matting brings 11%, 8%, 11%, and 10% relative improvement than the runner-up in terms of Mean Absolute Difference (MAD), Mean Squared Error (MSE), Gradient (Grad), and Connectivity (Conn) metrics in Figure 1a. Moreover, Figure 1b demonstrates 32%, 11%, 45%, 21%, and 5% relative improvement made by our SEMat trained by COCO-Matting on five datasets, including P3M, AIM, RW100, AM, and RWP636, compared with the second-best SmartMat (Ye et al., 2024).

## 2 RELATED WORKS

**Interactive Matting**. Trimap-based methods (Yao et al., 2024a; Hu et al., 2023) have demonstrated impressive accuracy in image matting. However, trimaps are hard to obtain in real-world scenarios. Consequently, several approaches have sought alternative auxiliary inputs, such as backgrounds (Lin et al., 2021), coarse segmentation maps (Yu et al., 2021), and interactive prompts (e.g., points, scribbles, and BBox) (Yang et al., 2022; 2020).

To eliminate the ambiguity of multiple objects in one image, UIM (Yang et al., 2022) introduces interactive prompts and decouples the matting into foreground segmentation and transparency prediction. However, it is trained and tested only on the synthetic dataset (Xu et al., 2017). Recently, MatAny (Yao et al., 2024b) and MAM (Li et al., 2024) leverage the robust generalization capabilities of pre-trained SAM (Kirillov et al., 2023) to transfer from interactive segmentation to interactive matting. MatAny employs a training-free two-stage pipeline: it creates trimaps by eroding the masks generated from SAM and then predicts alpha mattes with a trained trimap-based network (Yao et al., 2024a). In contrast, MAM trains a learnable mask-to-matte module that takes predicted mask and intermediate features as input. On the other hand, SmartMat (Ye et al., 2024) extracts candidate features from the DINOV2 pre-trained ViT (Oquab et al., 2024) based on both saliency and interactive information, achieving a unified approach for both automatic and interactive matting.

**Matting Datasets**. Matting, as opposed to segmentation, aims to predict the alpha mattes of target objects which are continuous transparency between 0 and 1 rather than binary masks. The majority of matting datasets are limited in size due to the high annotation cost, often consisting of only a few thousand or even just a few hundred foreground objects (Liu et al., 2021; Sun et al., 2021). For instance, the Composition-1k dataset (Xu et al., 2017) comprises 481 foreground instances, encompassing a variety of matting scenarios such as hair, fur, and semi-transparent objects. Considering the repetition of some foregrounds in the Composition-1k dataset (cropped patches from the same image), the Distinction-646 dataset (Qiao et al., 2020) is proposed, featuring 646 distinct foreground images. Focusing on specific categories, the AM-2K dataset (Li et al., 2022) includes 20 categories of animals, with 100 real-world images per category, showcasing diverse appearances and forms. To amass a sufficient amount of training data, each foreground in the above datasets is typically merged with 20 to 100 different backgrounds to synthesize an expanded dataset (Li et al., 2023a).

Although there exists natural human matting dataset P3M-10K (Li et al., 2021a) which contains approximately 10,000 annotations, it is designed for simple single-instance scenarios, where each image features only a single person as the primary subject. Training with the synthetic or simple natural datasets may prevent the network from learning complicated semantic cues and lead to inaccuracies in distinguishing foreground from background in complex real-world scenarios.

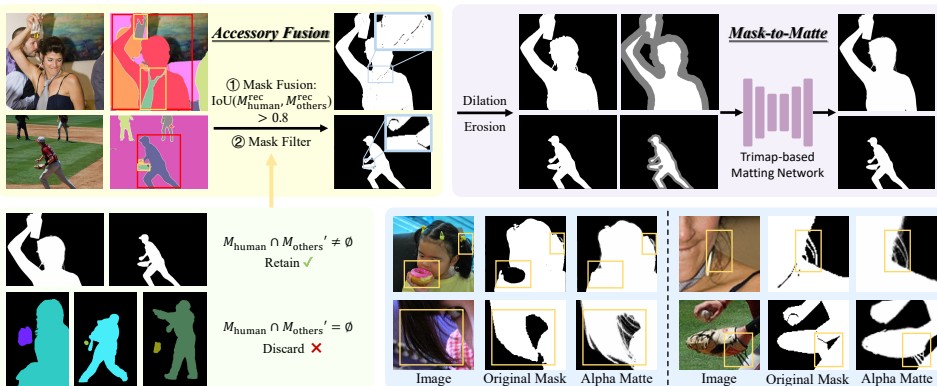

Figure 2: The construction of our proposed COCO-Matting dataset is divided into two parts: Accessory Fusion and Mask-to-Matte. Finally, a comparison of original masks and alpha mattes is shown.

## 3 COCO-MATTING DATASET

### 3.1 INVESTIGATION AND MOTIVATION

Interactive matting (Li et al., 2024; Ye et al., 2024) aims to predict the alpha matte of a specific object using a simple prompt, such as BBox. However, its training process typically requires labor-intensive and scarce alpha mattes. To mitigate this challenge, synthetic techniques are often employed to expand the training dataset by applying spatial transformations to foreground alpha mattes and overlaying them onto diverse background images. Unfortunately, training on synthetic data frequently fails to accurately distinguish between foreground and background in real-world and complex scenes. For example, as illustrated in Figure 1b (III), SmartMat (Ye et al., 2024), a SoTA interactive matting method, has insufficient semantic comprehensibility and struggles to correctly segment foreground objects in scenes with complex occlusions, such as wrongly predicting intersection regions when two people overlap.

### 3.2 DATASET CREATION

Inspired by the findings in Section 3.1, we aim to leverage a large-scale, real-world segmentation dataset to improve the network's semantic understanding. The COCO dataset (Lin et al., 2014), being one of the most representative and comprehensive benchmarks for segmentation, is a natural choice. However, due to the low-quality annotated masks in the original COCO dataset, we use the images from COCO combined with the refined annotations from COCONut (Deng et al., 2024) to construct our COCO-Matting dataset. Given the complex interactions between humans and their surrounding environments, we focus on humans as the primary subjects, which also has many real-world applications, such as background replacement in video conferences (Lin et al., 2021) and movie special effects production (Fielding, 2013).

Construction of the COCO-Matting dataset is fraught with two principal challenges. Firstly, the "human" category within the COCO dataset excludes accessories, such as hats, backpacks, or items being held, which are considered part human in matting tasks (Li et al., 2021a), thus creating a difference in label distribution. Secondly, annotations in COCO dataset are rough binary masks that lack precision and transparency, such as the delineation of hair strands. In response to these challenges, we devise a comprehensive pipeline including 1) Accessory Fusion 2) Mask-to-Matte as shown in Figure 2, and establish the COCO-Matting dataset.

**Accessory Fusion.** This step aims to merge masks that may belong to human accessories through the overlap rate between masks and thus aligns with the distribution of human annotations in the matting dataset. Specifically, given the instance segmentation masks for an image, we transform each binary mask $M \in \{0, 1\}$ into the form of minimum bounding rectangle $M^{\text{rec}}$. For "human" mask $M^{\text{rec}}_{\text{human}}$, we measure its IoU with every other mask $M^{\text{rec}}_{\text{others}}$, which belongs to pre-defined possible accessories like bags, bottles, or ties. When the calculated IoU value, denoted by $\text{IoU}(M^{\text{rec}}_{\text{human}}, M^{\text{rec}}_{\text{others}})$, surpasses a predefined threshold $\tau$, we proceed to integrate $M_{\text{others}}$ into $M_{\text{human}}$ which intuitively

means that $M_{\text{others}}$ is probably an accessory of $M_{\text{human}}$. In this work, $\tau$ is set to 0.8, empirically. As illustrated in Figure 2 (upper left), the woman in the red BBox corresponds to $M_{\text{human}}^{\text{rec}}$, while the tie and bottle in the yellow BBox represents those accessory objects $M_{\text{others}}^{\text{rec}}$ with an IoU exceeding the threshold when calculated with $M_{\text{human}}^{\text{rec}}$. Then, we integrate the masks of tie and bottle with the mask of woman, turning her into a woman wearing a tie and holding a bottle.

However, as shown in Figure 2 (lower left), the above fusion may incorrectly treat other objects like the green baseball mask as accessories. To counteract this, it is necessary to filter the mask $M_{\text{others}}$. We firstly apply a dilation operation (Yao et al., 2024b) with a small kernel size of 4 and acquire $M_{\text{others}}' = \text{Dilate}(M_{\text{others}}, 4)$ to expand the shape of $M_{\text{others}}$. Then, if the intersection of $M_{\text{human}}$ and $M_{\text{others}}'$ is empty, e.g., the baseball and the man holding the bat in Figure 2, we abandon this fusion. With the mask filter, our accessory fusion successfully incorporates items such as bottles, ties, and gloves into the "human" category and gets $M_{\text{fusion}}$, in accordance with the criteria of matting tasks.

**Mask-to-Matte.** Here we aim to convert binary coarse masks into continuous high-precision alpha mattes that match the matting annotations. Specifically, given a fused mask $M_{\text{fusion}}$, a straightforward way to obtain alpha mattes is to obtain the trimap by erosion and forward it to the trained trimap-based matting network. Nonetheless, after fusion as shown in the blue box in Figure 2, gaps between the masks of different objects are discernible, potentially impeding the generation of correct trimaps. To address this, we firstly obtain $M_{\text{fusion}}' = \text{Dilate}(M_{\text{fusion}}, 4)$ by a dilation operation, and then perform two erosion operations on the mask and gain

$$M_{\text{fusion}}^{\text{ero}}{}' = \text{Erode}(M_{\text{fusion}}', \omega) \ \text{ and } \ \tilde{M}_{\text{fusion}}^{\text{ero}}{}' = \text{Erode}(1 - M_{\text{fusion}}', \omega), \ \text{ where } \omega = \sqrt{\sum\nolimits_{i,j} M_{\text{fusion}}'}/\eta.$$

Here $\omega$ denotes an adaptive kernel size based on mask area and $\eta$ is the scale factor set to 12 in our experiments. Finally, we forward the following trimap $T(x)$ to the trained trimap-based network (Hu et al., 2023) and obtain the corresponding alpha mattes $A$ which is our target.

$$T(x) = \begin{cases} 1, & \text{if } M_{\text{fusion}}^{\text{ero}}{}'(x) = 1, \\ 0, & \text{if } \tilde{M}_{\text{fusion}}^{\text{ero}}{}'(x) = 1, \\ 0.5, & \text{otherwise.} \end{cases}$$

With the above two steps, we convert the original masks into alpha mattes with superior precision. As shown in the lower right part of Figure 2, our generated alpha mattes exhibit refined edge transitions, particularly enhancing intricate details such as hair strands and the contours of objects like baseball gloves. Table 1 highlights the advantages of our COCO-Matting dataset over previous datasets: 1) COCO-Matting includes a diverse and complex "human" class, with 38,251 alpha mattes—the largest of its kind. 2) Unlike P3M-10K which contains only a single human instance per image, our COCO-Matting features multiple human instances within complex scenes. 3) COCO-Matting presents natural scenes, as opposed to RefMatte which synthesizes multi-instance samples on arbitrary and disharmonious backgrounds.

Table 1: Comparison of our COCO-Matting with existing matting datasets. "Number" denotes number of alpha mattes. "Complex" and "Natural" refer to whether a complex scene contains multi-instance and whether the foreground is in the natural scene, respectively.

| Datasets | Number | Class | Complex | Natural |
|---|---|---|---|---|
| Comp.-1K (Xu et al., 2017) | 481 | Object | ✗ | ✗ |
| Dist-646 (Qiao et al., 2020) | 646 | Object | ✗ | ✗ |
| AM-2K (Li et al., 2022) | 2,000 | Animal | ✗ | ✓ |
| Human-2K (Liu et al., 2021) | 2,100 | Human | ✗ | ✗ |
| P3M-10K (Li et al., 2021a) | 10,421 | Human | ✗ | ✓ |
| RefMatte (Li et al., 2023b) | 13,181 | Human | ✓ | ✗ |
| COCO-Matting | 38,251 | Human | ✓ | ✓ |

## 4 SEMANTIC ENHANCED MATTING METHODOLOGY

Here we first introduce our proposed network architecture and then elaborate on its training loss.

### 4.1 SEMAT NETWORK

Our proposed SEMat builds on SAM (Kirillov et al., 2023), as illustrated in Figure 3. While previous works like MatAny (Yao et al., 2024b) and MAM (Li et al., 2024) are also based on SAM, they rely on intermediate features or coarse masks from a frozen SAM, using additional refinement networks

Figure 3: Given an image and a box prompt, the alpha matte is obtained by the process of our proposed Feature-Aligned Transformer and Matte-Aligned Decoder sequentially. Furthermore, combined with the traditional matting loss, the frozen SAM and trimap annotations are introduced to calculate the regularization and trimap loss during training.

to produce alpha mattes. However, these kinds of approaches face two significant challenges. **(1) Unaligned Features:** Matting demands precise attention to edge details and object transparency, which cannot be adequately captured by simply extracting features from a frozen SAM designed primarily for rough object segmentation. **(2) Unaligned Mattes:** The SAM decoder struggles to accurately segment objects specific to matting, such as smoke, nets, and silk as shown in (Sun et al., 2021), which are not common in typical segmentation tasks. This mismatch can severely degrade the performance of subsequent refinement stages.

To address the first challenge, we propose a Feature-Aligned Transformer, which fuses the prompt with image patches during the input stage through an additional patch embedding layer and integrates LoRA (Hu et al., 2022) into the ViT backbone. Accordingly, it can focus more on the prompt region while tuning itself to extract fine-grained and transparency features. For the second challenge, we introduce the Matte-Aligned Decoder, which adds several matting tokens and the matting adapter in SAM's mask decoder to better segment matting-specific objects. Additionally, a lightweight, UNet-inspired (Ronneberger et al., 2015) matting decoder is incorporated to further refine the results. Consequently, our decoder is able to segment matting-specific objects and convert coarse masks into high-precision mattes. Below, we elaborate on both of these solutions in detail.

**Feature-Aligned Transformer.** Since our goal is to perform matting for specific objects of interest, we use their BBox prompts as additional guidance to help the ViT backbone focus on learning the features of these target objects. This extra guidance is essential because, without it, ViT would extract generic features for segmentation, making it difficult to distinguish the foreground object and its transparency. So in Figure 3, we convert the BBox prompt into a binary mask and concatenate it with the input image to forward a learnable embedding layer. This process directs ViT to identify which objects are of interest, thereby aligning the extracted features with the matting task.

Additionally, we incorporate LoRA (Hu et al., 2022) into the ViT backbone for efficient fine-tuning, further aligning the feature extraction with the matting task. Moreover, by focusing on the features of the specific objects indicated by the BBox prompts, the ViT backbone can learn more fine-grained details, such as precise edges and transparency, which are crucial for accurate matting.

**Matte-Aligned Decoder.** To effectively segment matting-specific objects, we enhance the SAM mask decoder by introducing three specialized matting tokens and a matting adapter. This shares a similar spirit with HQ-SAM (Ke et al., 2024) which employs an additional learnable HQ-token to refine segmentation masks. As depicted in Figure 3, the SAM mask decoder augmented by matting adapter processes the extracted image features $\{f_i\}$ alongside a concatenation of BBox prompt tokens, SAM tokens from vanilla SAM, and our learnable matting tokens for generating mask features $f_M \in \mathbb{R}^{[H \times W, C]}$ and output tokens $t_O \in \mathbb{R}^{[4,C]}$. For further details, refer to Appendix A.1.

Next, we compute the logits $\hat{p} = t_O f_M \in \mathbb{R}^{[4, H \times W]}$, and split it into two parts, i.e., SAM logits $\hat{p}^{\text{SAM}} \in \mathbb{R}^{[1, H, W]}$ and matting logits $\hat{p}^{\text{Mat}} \in \mathbb{R}^{[3, H, W]}$. Among them, $\hat{p}^{\text{SAM}}$ denotes the original SAM mask prediction, crucial for constructing the regularization loss outlined in Section 4.2, while $\hat{p}^{\text{Mat}}$ represents three-channel course-grained matting mask prediction, and will be adopted to generate the trimap enriched with enhanced semantic knowledge and constructs the trimap loss in Section 4.2.

During training, our learnable matting tokens and matting adapter are meticulously fine-tuned to discern matting-specific objects, ensuring accurate segmentation and mask generation that align with matting requirements. Unlike HQ-SAM (Ke et al., 2024), which derives binary masks from a single HQ-token, we employ multiple matting tokens to generate a trimap that meticulously distinguishes between foreground, background, and unknown regions of an object. Additionally, $\hat{p}^{\text{SAM}}$ is preserved throughout the training process to maintain the pre-trained SAM's inherent priors.

However, due to memory constraints, SAM extracts features at a reduced resolution and generates logits downsampled by a factor of 4, leading to a loss of fine details. To address this, we propose a lightweight matting decoder that stacks residual blocks (He et al., 2016) in a UNet architecture (Ronneberger et al., 2015). The central concept is that the network takes the concatenation of the image and upsampled matting logits Upsample($\hat{p}^{\text{Mat}}$) as input. The U-shaped structure, augmented with skip connections, effectively retains high-resolution image details, allowing the decoder to recover fine-grained details within the matting logits and produce high-precision alpha mattes. More details of the matting decoder and trainable parameters in SEMat are presented in Appendix A.2 and A.3.

## 4.2 SEMat Training Objective

Existing interactive matting (Li et al., 2024) is trained by the end-to-end matting loss. We argue that utilizing only matting loss may corrupt the prior of the pre-trained model and produce erroneous foreground segmentation. Therefore, we propose a regularization loss to ensure the consistency of predictions between the frozen and learnable networks. In addition, to enhance the semantic guidance during mask generation, we propose a trimap prediction loss which encourages the matting logits extracted from the mask decoder to contain trimap-based semantic information.

**Regularization Loss.** Despite the introduction of real-world COCO-Matting data enhancing the semantic comprehension, synthetic training data inevitably undermine the robust feature representations of the pre-trained SAM. Driven by this insight, we introduce a novel regularization loss designed to maintain the original priors during training. Specifically, We integrate an additional frozen SAM that processes an image alongside the BBox prompt to yield SAM logits $p^{\text{SAM}} \in [0, 1]$. Then, we generate the label $y_{i,j}^{\text{SAM}} = \mathbf{1}_{p^{\text{SAM}} > 0.5}$ to supervise the predicted SAM logits $\hat{p}^{\text{SAM}}$:

$$\mathcal{L}_{\text{Reg}} = -\sum\nolimits_{i,j} y_{i,j}^{\text{SAM}} \log(\hat{p}_{i,j}^{\text{SAM}}) + (1 - y_{i,j}^{\text{SAM}}) \log(1 - \hat{p}_{i,j}^{\text{SAM}}).$$

This binary cross-entropy loss ensures that our model can retain the pre-trained knowledge essential for generalizing to real-world samples.

**Trimap Loss.** A trimap divides an image into foreground, background, and unknown regions. Given an image and its trimap, trimap-based matting network (Yao et al., 2024a) is tasked solely with predicting refined alpha mattes for the unknown regions, without distinguishing between foreground and background. It allows trimap-based methods to achieve higher accuracy. Building on this idea, we explore whether the benefits of the trimap's rich semantic information can be utilized without directly using it as input. Here we use the trimap annotations to supervise the matting logits with Gradient Harmonizing Mechanism (GHM) loss (Li et al., 2019):

$$\mathcal{L}_{\text{Trimap}} = -\frac{1}{\text{GD}(g_{i,j})} \sum\nolimits_{i,j} \log(\hat{p}_{i,j}^{\text{Mat}}),$$

where $\hat{p}_{i,j}^{\text{Mat}}$ denotes the confidence calculated with the trimap annotations at the point $(i, j)$. $\text{GD}(g_{i,j})$ denotes the density of gradient $g_{i,j} = y_{i,j}^{\text{Tri}} - p_{i,j}^{\text{Mat}}$ of the cross-entropy loss $(-\log p_{i,j}^{\text{Mat}})$, where $y_{i,j}^{\text{Tri}}$ is the ground-truth trimap. Intuitively, for too easy or too difficult samples, their corresponding density $\text{GD}(g_{i,j})$ are often large and thus have a small loss weight $1/\text{GD}(g_{i,j})$ (Li et al., 2019).

**Matting Loss.** We follow the loss in ViTMatte (Yao et al., 2024a), i.e., adopt the $\ell_1$ loss on known and unknown regions, gradient penalty loss (Dai et al., 2022) and laplacian loss (Hou & Liu, 2019):

$$\mathcal{L}_{\text{Mat}} = \|\alpha - \hat{\alpha}\|_{1,\mathcal{K}} + \|\alpha - \hat{\alpha}\|_{1,\mathcal{U}} + (\|\nabla\alpha - \nabla\hat{\alpha}\|_1 + \lambda \|\nabla\alpha\|_1) + \sum\nolimits_{k=1}^{5} 2^{k-1} \|L^k(\alpha) - L^k(\hat{\alpha})\|_1 ,$$

where $\hat{\alpha}$ is the predicted alpha mattes, $\alpha$ is the ground truth, $\mathcal{K}$ denotes the known region in trimap, $\mathcal{U}$ denotes the unknown region in trimap, and $L^k(\cdot)$ denotes the $k^{th}$ output of the laplacian pyramid.

Now we are ready to define the overall objective which is a combination of the three losses:

$$\mathcal{L} = \mathcal{L}_{\text{Mat}} + \lambda_{\text{R}} \mathcal{L}_{\text{Reg}} + \lambda_{\text{T}} \mathcal{L}_{\text{Trimap}}, \tag{1}$$

where $\lambda_{\text{R}}$ and $\lambda_{\text{T}}$ are weights to balance the loss.

Table 2: Quantitative results of our SEMat and other methods on six datasets, including P3M-500-NP, AIM-500, RefMatte-RW100, AM-2K, RWP636, and SIM. The best and second best are highlighted. "Impro." denotes the average relative improvement on the five metrics compared with the baseline MatAny.

| Method | Pretrained Backbone | P3M-500-NP (Li et al., 2021a) | | | | | | AIM-500 (Li et al., 2021b) | | | | | |
|---|---|---|---|---|---|---|---|---|---|---|---|---|---|
| | | SAD↓ | MSE↓ | MAD↓ | Grad↓ | Conn↓ | Impro.↑ | SAD↓ | MSE↓ | MAD↓ | Grad↓ | Conn↓ | Impro.↑ |
| SmartMat | DINOv2 | 5.01 | 0.0026 | 0.0070 | 8.11 | 3.66 | 58.9% | 11.19 | 0.0077 | 0.0152 | 14.48 | 6.28 | 56.2% |
| MatAny | SAM | 21.67 | 0.0243 | 0.0294 | 10.79 | 5.02 | - | 38.71 | 0.0428 | 0.0516 | 18.07 | 10.05 | - |
| MAM | | 7.54 | 0.0051 | 0.0104 | 6.35 | 4.10 | 53.7% | 14.12 | 0.0090 | 0.0187 | 10.43 | 7.74 | 54.3% |
| SEMat | | 3.91 | 0.0021 | 0.0054 | 5.00 | 2.98 | 69.8% | 11.46 | 0.0078 | 0.0154 | 7.76 | 5.92 | 64.1% |
| SEMat | HQ-SAM | 3.42 | 0.0016 | 0.0048 | 4.90 | 2.49 | 73.6% | 10.01 | 0.0061 | 0.0134 | 7.85 | 5.77 | 66.6% |
| SEMat | SAM2 | 4.28 | 0.0027 | 0.0060 | 5.06 | 3.04 | 68.3% | 10.52 | 0.0069 | 0.0142 | 7.68 | 5.95 | 65.5% |
| | | RefMatte-RW100 (Li et al., 2023b) | | | | | | AM-2K (Li et al., 2022) | | | | | |
| SmartMat | DINOv2 | 11.80 | 0.0143 | 0.0168 | 11.09 | 1.64 | -17.4% | 6.59 | 0.0040 | 0.0091 | 10.28 | 4.23 | -3.2% |
| MatAny | SAM | 9.19 | 0.0107 | 0.0132 | 9.89 | 1.92 | - | 7.19 | 0.0046 | 0.0100 | 7.06 | 4.20 | - |
| MAM | | 12.39 | 0.0130 | 0.0178 | 8.91 | 2.34 | -20.6% | 6.11 | 0.0028 | 0.0085 | 6.09 | 4.13 | 16.9% |
| SEMat | | 6.73 | 0.0074 | 0.0097 | 6.13 | 1.58 | 28.0% | 5.13 | 0.0026 | 0.0071 | 4.78 | 3.46 | 30.2% |
| SEMat | HQ-SAM | 6.51 | 0.0074 | 0.0094 | 5.89 | 1.29 | 32.4% | 5.23 | 0.0026 | 0.0073 | 4.83 | 3.41 | 29.6% |
| SEMat | SAM2 | 5.08 | 0.0054 | 0.0073 | 5.56 | 1.23 | 43.7% | 5.20 | 0.0027 | 0.0072 | 4.71 | 3.45 | 29.6% |
| | | RWP-636 (Yu et al., 2021) | | | | | | SIM (Sun et al., 2021) | | | | | |
| SmartMat | DINOv2 | 17.32 | 0.0102 | 0.0210 | 35.06 | 13.33 | 50.6% | 51.16 | 0.0448 | 0.0689 | 36.29 | 22.97 | 52.7% |
| MatAny | SAM | 55.43 | 0.0566 | 0.0656 | 45.06 | 15.18 | - | 118.07 | 0.1273 | 0.1527 | 77.86 | 34.62 | - |
| MAM | | 25.68 | 0.0161 | 0.0302 | 32.42 | 16.51 | 39.7% | 56.76 | 0.0418 | 0.0754 | 53.12 | 29.66 | 43.2% |
| SEMat | | 16.95 | 0.0102 | 0.0204 | 31.66 | 13.24 | 52.6% | 23.16 | 0.0103 | 0.0310 | 18.45 | 17.69 | 75.4% |
| SEMat | HQ-SAM | 16.53 | 0.0095 | 0.0199 | 31.64 | 13.04 | 53.4% | 23.28 | 0.0105 | 0.0312 | 17.73 | 17.31 | 75.8% |
| SEMat | SAM2 | 15.79 | 0.0093 | 0.0191 | 30.96 | 12.39 | 55.1% | 20.51 | 0.0072 | 0.0269 | 17.25 | 16.68 | 77.8% |

## 5 EXPERIMENTS

**Datasets and Metrics.** We employ the Distinction-646 (Qiao et al., 2020), AM-2K (Li et al., 2022), Composition-1k (Xu et al., 2017) adopted in SmartMat (Ye et al., 2024), and our proposed COCO-Matting datasets for training. Evaluation is conducted across various benchmarks, encompassing P3M-500-NP (Li et al., 2021a), RWP636 (Yu et al., 2021), and RefMatte-RW100 (Li et al., 2023b) for human matting, AM-2K (Li et al., 2022) for animal matting, AIM-500 (Li et al., 2021b) for object matting, and SIM (Sun et al., 2021) for synthetic image matting. We leverage five standard metrics for evaluation: Sum of Absolute Difference (SAD), Mean Squared Error (MSE), Mean Absolute Difference (MAD), Gradient (Grad), and Connectivity (Conn) (Rhemann et al., 2009), where a lower value indicates superior performance. Additionally, we assess our approach on the HIM2K human instance matting dataset (Sun et al., 2022a) with the Instance Matting Quality (IMQ), where a higher value signifies a more favorable outcome. For fair comparison, all methods take images resized proportionally to 1024 resolution as input in evaluation.

**Training Details.** We initialize our network with SAM (Kirillov et al., 2023), HQ-SAM (Kirillov et al., 2023), and SAM2 (Ravi et al., 2024), respectively. The learnable modules are trained with 60k iterations and a batch size of 2 on one Nvidia L40S GPU. AdamW is chosen as the optimizer with a learning rate of 5e-5. The rank is set to 16 in LoRA. $\lambda_R$ and $\lambda_T$ are set to 0.2 and 0.05.

### 5.1 RESULTS ON INTERACTIVE MATTING

**Comparison with SoTAs.** Here we train our proposed SEMat built upon three different pre-trained backbones (SAM, HQ-SAM, and SAM2) on our COCO-Matting dataset, and compare them with SoTAs such as MatAny (Yao et al., 2024b), MAM (Li et al., 2024), and SmartMat (Ye et al., 2024).

Table 2 summarizes the quantitative results on six test datasets, and shows the superior performance of our proposed SEMat in all datasets as evidenced by the average relative improvement "Improv." on widely used five metrics. Specifically, **(a)** building upon the same SAM, SEMat improves the baseline MatAny and MAM by significant overall relative improvement on six datasets. **(b)** Based on HQ-SAM and SAM2, our SEMat can further improve its SAM based counterpart. **(c)** Compared

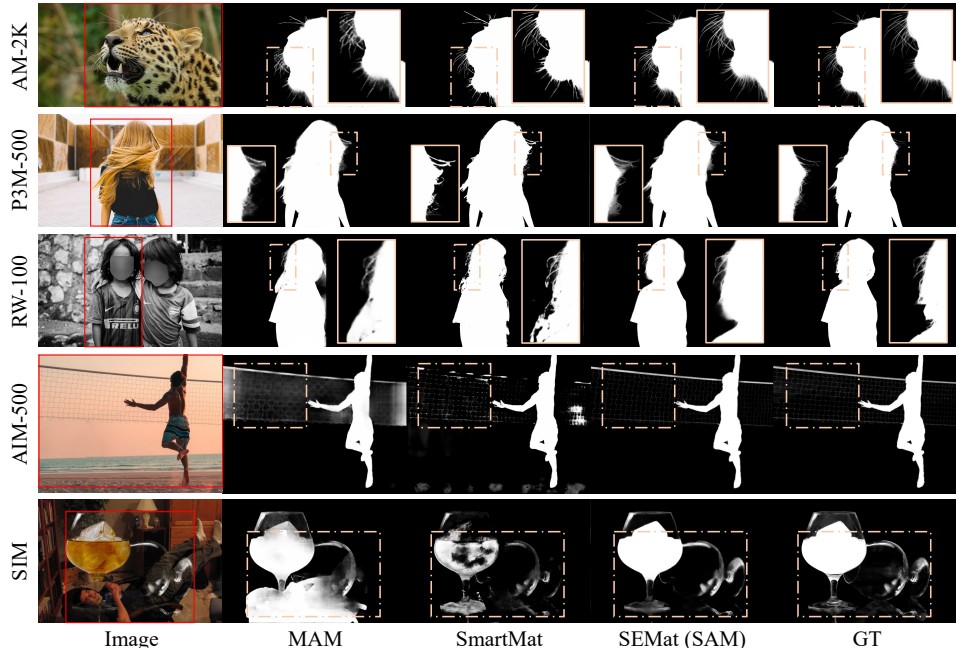

Figure 4: Qualitative matting results of MAM (Li et al., 2024), SmartMat (Ye et al., 2024), and our SEMat (SAM) on different datasets. See more matting results in Appendix A.4.

with previous SoTA SmartMat, our three SEMat versions always surpass it by a big margin. For instance, our HQ-SAM based SEMat makes 9.4%, 9.3%, 61.1%, 32.8%, 2.8%, and 13.1% relative improvement on the six datasets, which highlights the superiority and robustness of our SEMat.

Figure 4 shows a qualitative assessment. Our SAM-based SEMat showcases superior detail preservation, such as animal and human hair in the first and second row. Meanwhile, the enhanced semantic understanding of our methods is evident with occlusions or similar color distributions between the foreground and background in the third row. For multi-instance matting in the fourth row, our SEMat is able to predict both net and human mattes. Also, our method achieves better performance in the perception of transparent objects in the fifth row. See more qualitative results in Appendix A.4.

**Comparison of Human Instance Matting So-TAs.** In interactive matting, BBox prompts are obtained from the alpha matte annotations, which are not available in instance matting. Therefore, we integrate the object detection network Grounding DINO (Liu et al., 2023) for interactive matting methods, which outputs BBox for all "human" classes in each testing image. As illustrated in Table 3, compared to the dedicated instance matting method InstMatt (Sun et al., 2022b), previous interactive matting methods are not ideal for instance distinguish and have a poor performance.

Table 3: Results of different matting methods on the human instance matting HIM2K dataset.

| Method | HIM2K (Sun et al., 2022a) | | | | Impro.↑ |
|---|---|---|---|---|---|
| | IMQ$_{mad}$↑ | IMQ$_{mse}$↑ | IMQ$_{grad}$↑ | IMQ$_{conn}$↑ | |
| MAM | 53.99 | 67.17 | 62.46 | 56.11 | - |
| MatAny | 62.08 | 73.24 | 61.02 | 65.62 | 9.7% |
| SmartMat | 66.29 | 75.09 | 60.27 | 67.93 | 13.0% |
| InstMatt | 71.06 | 82.99 | 74.92 | 73.39 | 26.5% |
| SEMat (SAM) | 76.25 | 88.89 | 75.28 | 80.33 | 34.3% |
| SEMat (HQ-SAM) | 76.67 | 89.04 | 75.76 | 80.76 | 34.9% |
| SEMat (SAM2) | 77.32 | 89.70 | 76.31 | 81.52 | 36.1% |

Our three SEMat versions achieve superiority in handling the complexities of human instance matting, and respectively show 5.8%, 7.4%, and 9.6% average improvement over the SoTA InstMatt. Figure 5 in Appendix A.4 provides a clearer visualized comparison. Our SEMat exhibits exceptional semantics understanding and high-quality edge matting in multi-instance samples.

**Performance Improvement Analysis.** In Table 4, we independently investigate the performance improvement made by our COCO-Matting dataset and SEMat framework. Specifically, we independently combine Distinction-646 (Qiao et al., 2020), AM-2K (Li et al., 2022), Composition-1k (Xu et al., 2017) datasets with P3M-10K (Li et al., 2021a) or RefMatte (Li et al., 2023b) or COCO-Matting dataset for training MAM and our SAM-based SEMat methods. **(a)** By comparison, one can observe that with the same matting method, e.g., MAM or SEMat, our COCO-Matting dataset always guarantees much higher overall performance improvement, e.g., making 31.5% and 16.8%

Table 4: Benefits of different datasets on SAM-based MAM and our SEMat evaluated with HIM2K dataset. "Impro." denotes the average relative improvement on the four metrics compared with the baseline "+ P3M-10K ".

| Method | Training Dataset | HIM2K (Sun et al., 2022a) | | | | Impro.↑ |
|---|---|---|---|---|---|---|
| | | IMQ$_{mad}$↑ | IMQ$_{mse}$↑ | IMQ$_{grad}$↑ | IMQ$_{conn}$↑ | |
| MAM | + P3M-10K | 39.11 | 46.56 | 47.80 | 40.51 | - |
| | + RefMatte | 59.58 | 72.62 | 64.98 | 61.80 | 49.2% |
| | + COCO-Mat. | 75.23 | 87.04 | 72.50 | 77.65 | 80.7% |
| SEMat | + P3M-10K | 68.62 | 80.56 | 72.03 | 72.46 | 69.5% |
| | + RefMatte | 69.20 | 81.17 | 70.29 | 72.83 | 69.5% |
| | + COCO-Mat. | 76.67 | 89.04 | 75.76 | 80.76 | 86.3% |

Table 5: Effect investigation of hyperparameters $\lambda_R$ and $\lambda_T$ in the training loss (1). When adjusting one hyperparameter, another one uses the value in the gray background and always remains unchanged.

| $\lambda_T$ | 0.01 | 0.025 | 0.05 | 0.075 | 0.1 |
|---|---|---|---|---|---|
| Avg. SAD↓ | 6.92 | 6.68 | 6.29 | 6.61 | 6.63 |

| $\lambda_R$ | 0.1 | 0.15 | 0.2 | 0.25 | 0.30 |
|---|---|---|---|---|---|
| Avg. SAD↓ | 6.68 | 6.45 | 6.29 | 6.70 | 6.64 |

Table 6: Ablation study of different components in dataset, network, and training, evaluated with SAD metrics ↓ on four datasets. "Impro." denotes the average relative improvement on the "Avg." (average) metric compared with the baseline.

(a) COCO Mask

| | P3M | AIM | RW100 | AM | Avg.↓ | Impro.↑ |
|---|---|---|---|---|---|---|
| Baseline | 27.80 | 32.57 | 42.11 | 17.15 | 29.91 | - |
| + COCO Mask | 14.80 | 32.02 | 35.62 | 23.91 | 26.59 | 11.1% |

(b) Network

| | P3M | AIM | RW100 | AM | Avg.↓ | Impro.↑ |
|---|---|---|---|---|---|---|
| + LoRA | 9.89 | 17.43 | 12.50 | 7.98 | 11.95 | 60.0% |
| + Mat. Tok.&Ada. | 10.69 | 15.64 | 10.50 | 6.47 | 10.83 | 63.8% |
| + Prom. Enhan. | 9.78 | 16.12 | 9.26 | 5.99 | 10.29 | 65.6% |

(c) Dataset

| | P3M | AIM | RW100 | AM | Avg.↓ | Impro.↑ |
|---|---|---|---|---|---|---|
| + Acc. Fusion | 6.52 | 15.48 | 9.01 | 6.80 | 9.45 | 68.4% |
| + Mask-to-Mat. | 3.76 | 11.37 | 7.09 | 5.23 | 6.86 | 77.1% |

(d) Training

| | P3M | AIM | RW100 | AM | Avg.↓ | Impro.↑ |
|---|---|---|---|---|---|---|
| + Reg. Loss | 4.05 | 10.85 | 6.70 | 5.06 | 6.67 | 77.7% |
| + Trimap Loss | 3.42 | 10.01 | 6.51 | 5.23 | 6.29 | 79.0% |

overall improvement than the runner-up on MAM or SEMat, respectively. The superiority of our COCO-Matting can be attributed to its rich and diverse set of human instance-level alpha mattes which yields better generalization and robustness on complex and varied scenarios in HIM2K. **(b)** Building upon the same training datasets and the same SAM backbone, SEMat always surpasses MAM across the three settings, e.g., 45.1% overall improvement on the RefMatte-RW100 dataset.

## 5.2 ABLATION STUDY

Table 5 shows the effects of two hyperparameters $\lambda_T$ and $\lambda_R$ in the training loss (refer to Equation 1). One can observe that our method is generally robust to these two hyperparameters.

Next, we conduct four ablation studies in Table 6. The *baseline* in Table 6 denotes fine-tuning only a learnable matting decoder on the synthetic datasets. **(a) COCO Mask.** When integrating with the original COCO masks, the performance improves slightly due to simply taking the coarse masks as alpha matte annotations. **(b) Network.** By applying feature-aligned transformer and matte-aligned decoder, i.e., LoRA, prompt enhancement, matting token and adapter, our model demonstrates a superior extraction of matting-specific features. A substantial reduction in average SAD from 26.59 to 10.29 highlights the benefits of tailored network adjustments. **(c) Dataset.** Integrating with accessory fusion, SAD on the P3M human dataset has a significant reduction. Then, mask-to-matte provides fine-grained annotations that further reduce prediction errors. Contributions on our COCO-Matting help reduce the average SAD of -3.43. It is worth noting that although our COCO-Matting is human-centred, it also has a significant improvement on the AIM dataset including various objects. **(d) Training.** Our proposed regularization loss helps retain the pre-trained model's knowledge while adapting to matting data, leading to a balanced and robust performance. The final model with the addition of trimap loss exhibits the best performance, with the smallest average SAD of 6.29.

## 6 CONCLUSION

In this paper, we propose the COCO-Matting dataset and SEMat framework to revamp training datasets, network architecture, and training objectives. Solving the problem of inappropriate synthetic training data, unaligned features and mattes from a frozen SAM, and end-to-end matting loss lacking generalization, we greatly improve the interactive matting performance on diverse datasets. **Limitations:** Our method heavily relies on the pre-trained SAM, and its limitations may also affect our model's performance. For instance, while SAM is trained on large-scale data and effectively segments common objects, it struggles with rare objects, possibly limiting our model's capabilities.

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

## A APPENDIX

### A.1 DETAILS OF THE SAM MASK DECODER AND MATTING ADAPTER

---

**Algorithm 1** Details of the SAM Mask Decoder and Matting Adapter

---

**Require:** Shallow image features $f_0$, Deep image features $f_3$, Prompt tokens $t_P$, SAM tokens $t_{SAM}$, Matting tokens $t_{Mat}$

1: Obtain $t_{Input} = \text{Concat}(t_P, t_{SAM}, t_{Mat})$
2: **for** $n = 1$ **to** $2$ **do**
3:    $t_{Input} \mathrel{+}= \text{SelfAttn}(t_{Input})$
4:    $t_{Input} \mathrel{+}= \text{CrossAttn}(q = t_{Input}, k = f_3, v = f_3)$
5:    $t_{Input} \mathrel{+}= \text{MLP}(t_{Input})$
6:    $f_3 \mathrel{+}= \text{CrossAttn}(q = f_3, k = t_{Input}, v = t_{Input})$
7: **end for**
8: $t_{Output} = t_{Input} + \text{CrossAttn}(q = t_{Input}, k = f_3, v = f_3)$
9: $t_{Output}^{SAM}, t_{Output}^{Mat} = \text{Split}(t_{Output})$
10: $f_{Mask}^{SAM} = \text{UpsampledConv}(f_3)$
11: $f_{Mask}^{Mat} = \text{Conv}(f_{Mask}^{SAM}) + \text{UpsampledConv}(f_0) + \text{UpsampledConv}(f_3)$
12: $\hat{p}^{SAM} = \text{MLP}(t_{Output}^{SAM}) \cdot f_{Mask}^{SAM}$
13: $\hat{p}^{Mat} = \text{MLP}(t_{Output}^{Mat}) \cdot f_{Mask}^{Mat}$

---

In Algorithm 1, we elaborate on the forward details of the SAM mask decoder and matting adapter. The activation function and normalization are omitted for brevity. Learnable parameters are highlighted.

### A.2 DETAILS OF THE MATTING DECODER

Our matting decoder with the UNet shape takes the concatenation of the image and upsampled matting logits $\text{Upsample}(\hat{p}^{Mat})$ as input and extracts matting features through four downsampling residual blocks, progressively reducing the resolution to one-eighth of the original. Subsequently, intermediate features $\{f_i\}$ extracted from the ViT backbone are fused with the matting features through a residual block at the bottom of matting decoder. Lastly, four upsampling residual blocks with skip connections restore the features to the original resolution, yielding the alpha mattes.

### A.3 TRAINABLE PARAMETERS

Table 7: Overview of the trainable parameters (millions) distribution across the proposed SEMat.

| Component | SEMat (SAM & HQ-SAM) | SEMat (SAM2) |
|---|---|---|
| Prompt Enhancement | 1.0 M | 0.03 M |
| LoRA | 2.4 M | 2.7 M |
| Matting Tokens & Adapter | 1.3 M | 0.5 M |
| Matting Decoder | 13.6 M | 14.6 M |
| **Total** | **18.3 M** | **17.8 M** |

Table 7 presents a detailed account of the distribution of trainable parameters within the proposed SEMat of different versions. The table delineates four key components in our proposed feature aligned transformer and matte aligned decoder, each accompanied by the respective number of trainable parameters in millions. The total trainable parameters of 18.2 or 17.8 million are added to the pre-trained SAM (Kirillov et al., 2023; Ke et al., 2024; Ravi et al., 2024) to construct our SEMat.

### A.4 QUANTITATIVE RESULTS

In the subsequent part, we present a detailed quantitative analysis comparing the performance of InstMatt (Sun et al., 2022b), SmartMat (Ye et al., 2024), our novel SEMat (HQ-SAM), and SEMat (SAM2) on various benchmark datasets, including HIM-2K (Sun et al., 2022a), P3M-500-NP (Li et al., 2021a), RefMatte-RW100 (Li et al., 2023b), RWP636 (Yu et al., 2021), AIM-500 (Li et al., 2021b), AM-2K (Li et al., 2022) and SIM (Sun et al., 2021).

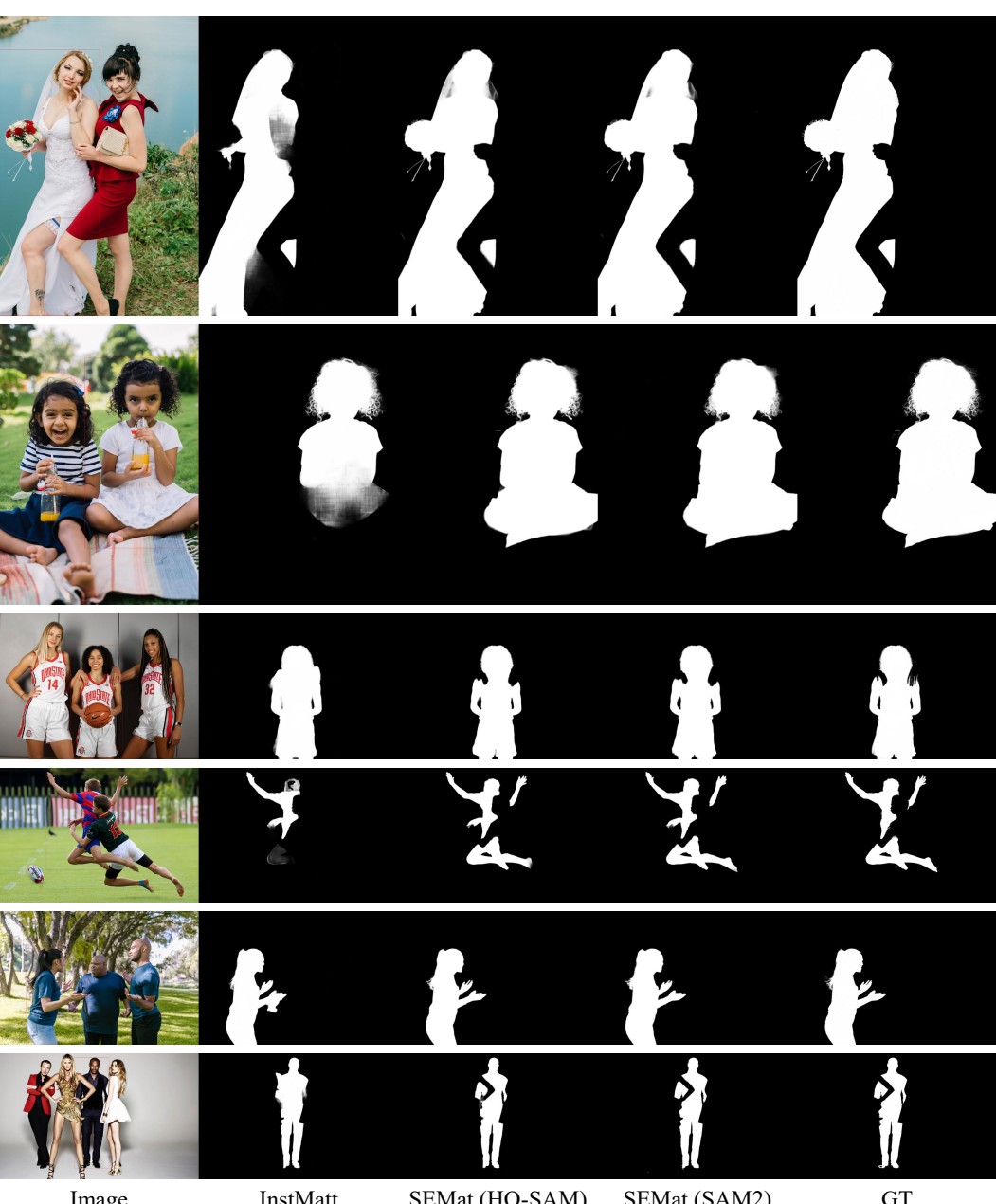

| Image | InstMatt | SEMat (HQ-SAM) | SEMat (SAM2) | GT |

Figure 5: Qualitative matting results on the HIM-2K dataset (Sun et al., 2022a) with InstMatt (Sun et al., 2022b), SEMat (HQ-SAM) and SEMat (SAM2).

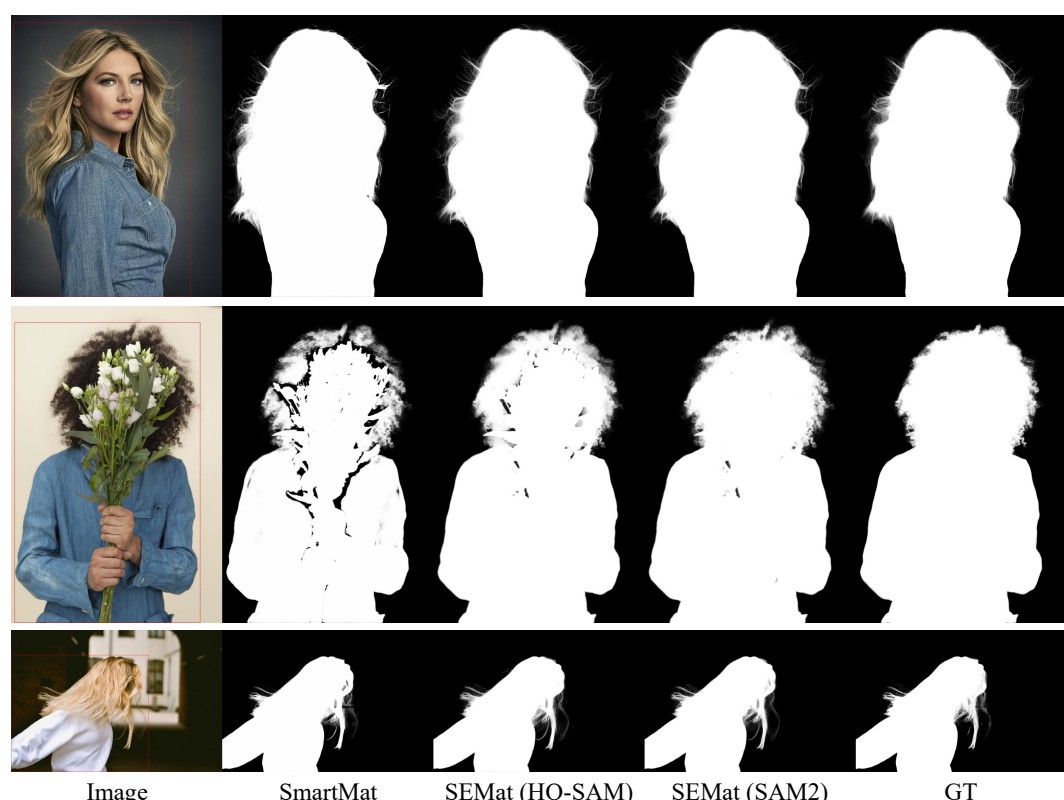

Figure 6: Qualitative matting results on the P3M-500-NP dataset (Li et al., 2021a) with Smart-Mat (Ye et al., 2024), SEMat (HQ-SAM) and SEMat (SAM2).

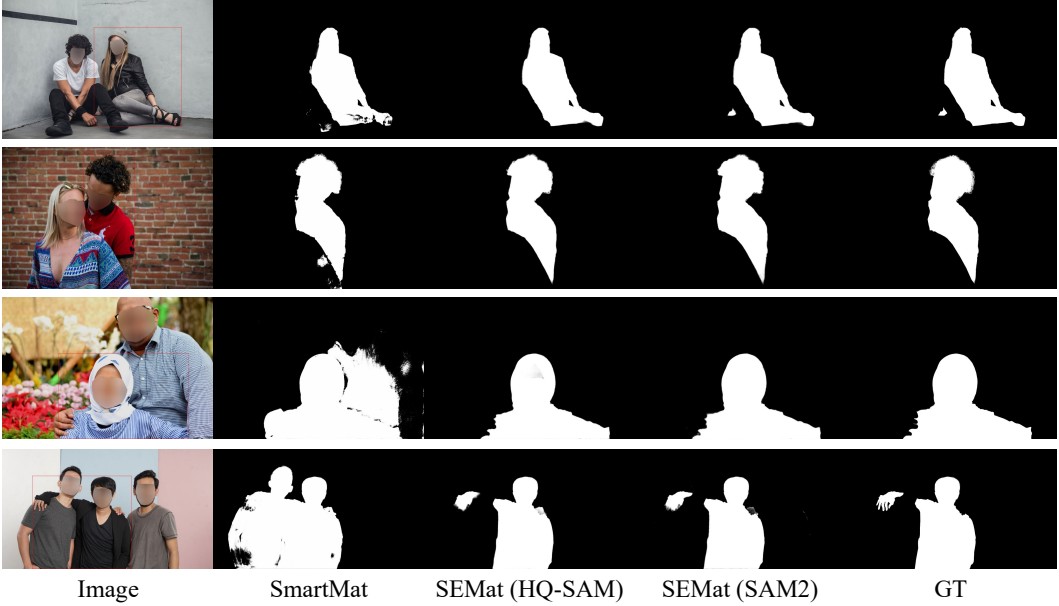

Figure 7: Qualitative matting results on the RefMatte-RW100 dataset (Li et al., 2023b) with Smart-Mat (Ye et al., 2024), SEMat (HQ-SAM) and SEMat (SAM2).

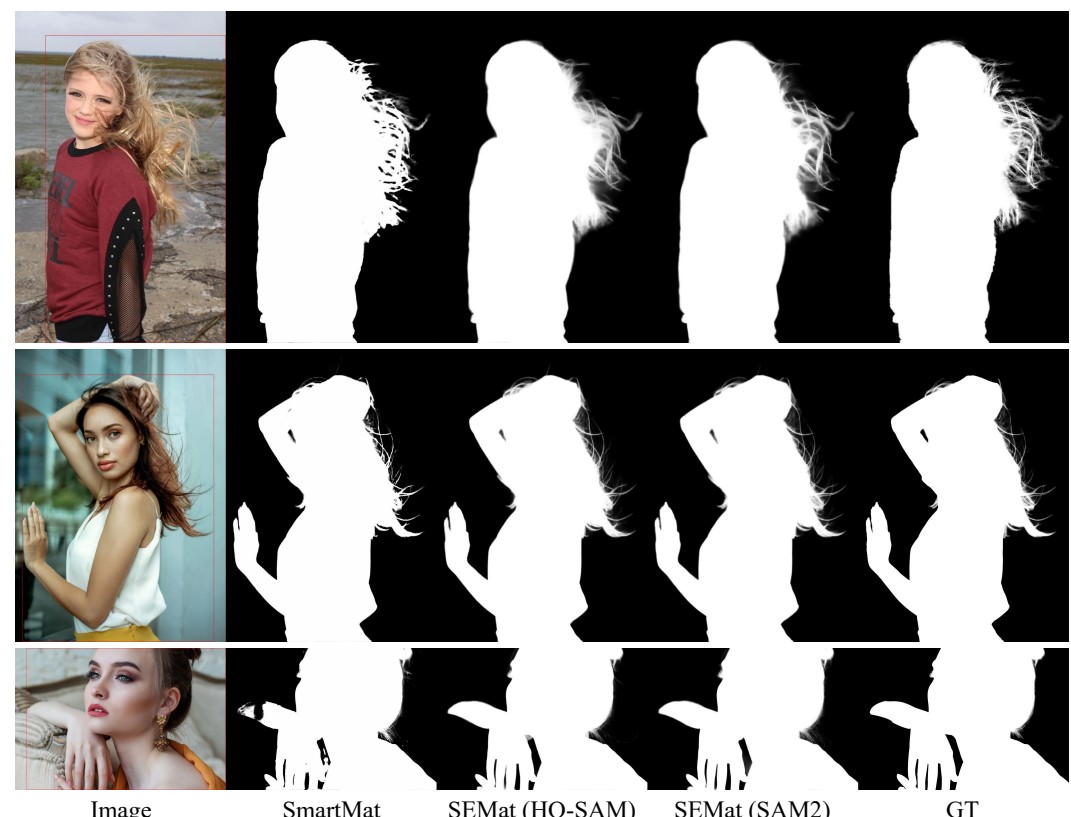

Figure 8: Qualitative matting results on the RWP-636 dataset (Yu et al., 2021) with SmartMat (Ye et al., 2024), SEMat (HQ-SAM) and SEMat (SAM2).

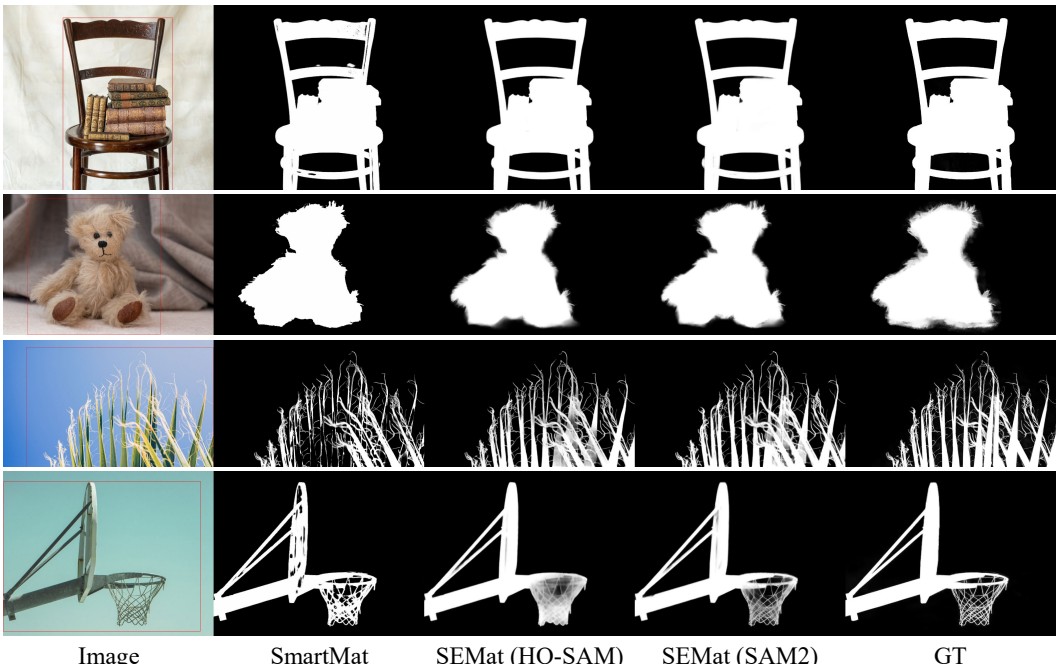

Figure 9: Qualitative matting results on the AIM-500 dataset (Li et al., 2021b) with SmartMat (Ye et al., 2024), SEMat (HQ-SAM) and SEMat (SAM2).

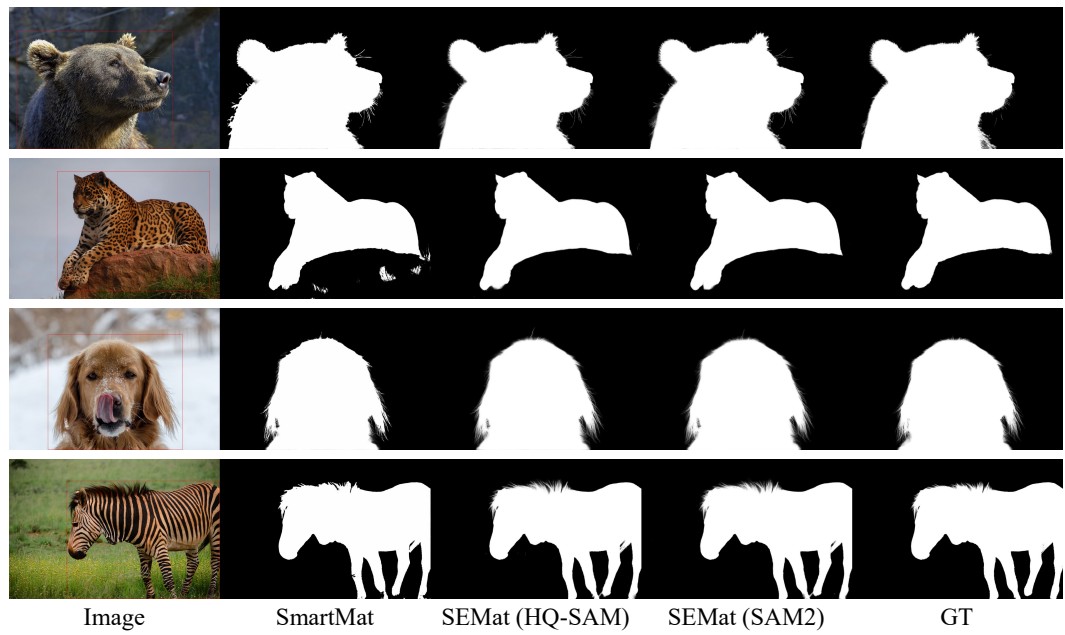

Figure 10: Qualitative matting results on the AM-2K dataset (Li et al., 2022) with SmartMat (Ye et al., 2024), SEMat (HQ-SAM) and SEMat (SAM2).

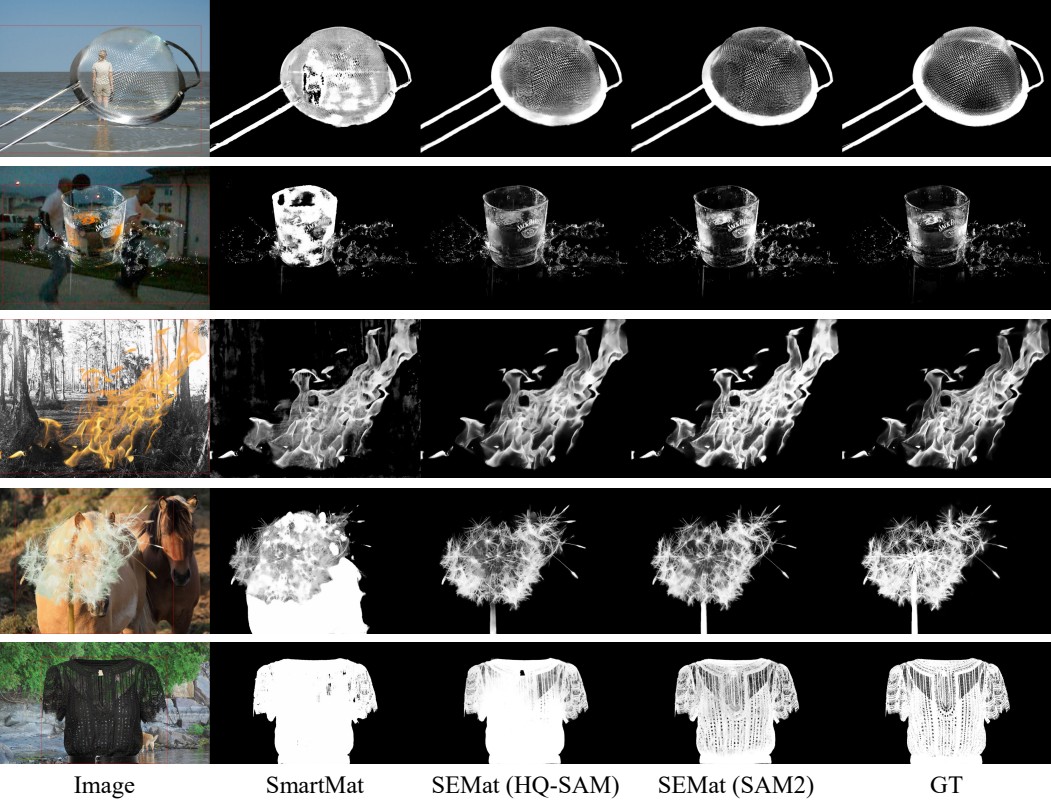

Figure 11: Qualitative matting results on the SIM dataset (Sun et al., 2021) with SmartMat (Ye et al., 2024), SEMat (HQ-SAM) and SEMat (SAM2).

