# OpenReview forum: "Towards Natural Image Matting in the Wild via Real-Scenario Prior"
_ICLR.cc/2025/Conference — ICLR 2025 Conference Withdrawn Submission_

### Official Review · Reviewer_uCWa · 2024-10-16

**Soundness:** 3
**Presentation:** 3
**Contribution:** 2
**Rating:** 6
**Confidence:** 4

**Summary:**

The paper presents the COCO-Matting dataset with 38,251 human alpha mattes for better generalization in natural scenes and the SEMat framework, which improves matting accuracy through a Feature-Aligned Transformer, Matte-Aligned Decoder, and enhanced loss functions. SEMat outperforms state-of-the-art methods in complex real-world scenarios.

**Strengths:**

1. The paper propose a pipeline to construct a new matting dataset based on a dataset of image segmentation. In this way, the matting model can be trained on a big and real dataset.
2. I think it is a highlight to predict trimap from SAM. Because a precise trimap is necessary for the matting decoder.

**Weaknesses:**

1. Novelty is not enough. Even though some tricks are used to construct a dataset, the matting model do not contain enough novel method.
2. The intro about matting dataset missed some existing datasets, training set and test set. For instance, RefMatte, ICM-57 and so on.
3. The ablation about the trimap loss is not clear. If you do not use the loss, what is used in your matting decoder to replace the trimap? In fact, I am concerned about the contribution of predicting trimap from SAM.

**Questions:**

Will the COCO-Matting be released?

---

### Official Review · Reviewer_1BbX · 2024-10-18

**Soundness:** 3
**Presentation:** 3
**Contribution:** 3
**Rating:** 5
**Confidence:** 4

**Summary:**

This paper improves interactive matting through constructing a new matting dataset based on COCO, and a modified version of SAM as the network. The dataset is processed in a reasonable way, and is surely a natural development for interactive matting. Also, the proposed SEMat is superior to other interactive methods. Extensive experiments are conducted to validate the effectiveness of proposed methods and datasets.

**Strengths:**

+The authors propose COCO-Matting, which is a valuable dataset for interactive matting. The results show that with COCO-Matting, which contains lots of natural images instead of synthetic images.

+The paper is well organized and presented, and the figures are also clear.

+The performance of SEMat is impressive, topping all the mentioned datasets.

**Weaknesses:**

Despite the strong performance and valuable datasets, I still have some concerns.

1.	Almost all of current matting datasets are manually labeled, but the authors just use trimap-based matting models to label it. So, whether the label can be called ground-truth is a question to be discussed. Is there any data selecting to verify the correctness of the dataset?

2.	From the results, the update of SAM version can introduce the improvement, so I wonder the result if MatAny is equipped with SAM2. Also, the results of original SAM/HQ-SAM/SAM2 is recommended to illustrate at the table.

3.	The authors provide the trainable parameters, but maybe the inference speed is also needed.

4.	Insufficient ablation study of network design. It will reveal more understanding if the authors can conduct ablation study to show why the lora and matting adapter is needed.

**Questions:**

Please see the weakness part. If my concerns are well addressed, I will lift my rating.

---

### Official Review · Reviewer_4H1Q · 2024-10-31

**Soundness:** 2
**Presentation:** 2
**Contribution:** 2
**Rating:** 5
**Confidence:** 5

**Summary:**

There are two major contritions for this paper.
1. Authors propose a new matting dataet that contains large-scale real-world images with matting level annotation.
2. Similar to other methods, authors also leverage the pre-trained frozen SAM and also frozen VIT feature for better matting results.

**Strengths:**

Authors propose a new large-scale dataset for matting but I felt there are more concerns on that.
Please see weakness.

**Weaknesses:**

1. The major contributions for this paper is the dataset. However, I have a very big concern on this part.

The way how authors generate the final 'high-quality' alpha matte is using a off-the-shelf method ."Finally,we forwardt he following trimapT(x) to the trained trimap-based network(Hu etal.,2023) and treat it as target.

My question is that since the GT is generated via another trimap-based matting method, meaning that's the highest quality you could get. Then, what's the purpose of your work ?  If I want to get the high-quality matted mask, I can just use SAM with some refinement method to get high-quality instance mask, and then generated the T(x) using Line 236 and Line 241-244, then I can get better quality that your method.

In this way, why I need authors method ?


2. I don't see any results authors compared with (Hu etal.,2023) which is used to generate the GT for the training dataset.  You can easily use your way to generate trimap and then run (Hu etal.,2023).

3. Performance Improvement Analysis section is the most important part in this paper especially to show the effectiveness of the proposed dataset. However, there is no summary of what is (a) (b) (c) )(d) means from the table. And there is almost no explanation  on that part in the main paper.

4. Since the major novelty of this paper is the contributed dataset, Authors should do a comprehensive analysis on the diversity of dataset and comprehensive experiments why this dataset is helpful.  Using your dataset and your new model to compare with a lot of other methods that trained on other dataset cannot tell what's the advantage of your dataset (basically your major novelty)

5. Authors should show results on green background to see how good your matting is. Showing mask only cannot tell much difference. Especially sharper mask or more matted hair doesn't mean your model is correct.  This could only be seen from the green background composition.

6.

**Questions:**

See weakness.  With all these weakness, it is hard to convince myself to accept this paper

---

### Official Review · Reviewer_iApZ · 2024-11-01

**Soundness:** 2
**Presentation:** 3
**Contribution:** 2
**Rating:** 3
**Confidence:** 5

**Summary:**

This paper attempts to address the poor generalization of SAM-based interactive image matting in reality. To this end, a COCO-Matting dataset is collected and annotated, containing 38,251 human instance-level alpha mattes in complex natural scenarios. Technically, the proposed SEMat enhances the SAM architecture by introducing the feature-aligned transformer and the matte-aligned decoder to extract fine-grained features and high-precision mattes, respectively. The regularization and trimap loss are proposed to preserve the semantic prior knowledge of the pre-trained SAM. Experiments across seven datasets show the superiority of the SEMat in the existing interactive natural human image matting.

**Strengths:**

+ Good motivation to construct a large-scale real-world human image matting dataset featuring complex scenes, annotated with human instance-level alpha mattes.
+ Feasible attempts for designing a matting-adapted SAM-based framework, with aligned network architecture and training losses. The framework seems effective.
+ Extensive experiments on seven datasets.
+ The paper is easy to follow.

**Weaknesses:**

i) Dataset
- Inaccurate annotations. The motivation to create the COCO-Matting is good, considering the natural and complex scenes, but I have two major concerns about the generation procedures of the annotations. First, I wonder whether the ‘Accessory Fusion’ step would still lead to cases where objects that do not correspond to the target instance are retained. For example, will the cup be retained if another human instance holds a cup that blocks part of the current instance's body? Also, in the lower left of Figure 2, if the baseball and the human instance are close enough, with a distance smaller than the dilation kernel, will the baseball be retained? Overall, I feel the fusion process is too heuristic and empirical, which may inevitably lead to uncertain annotation errors. I suggest the authors consider including manual post-screening to correct any potential errors. Another concern is in the ‘Mask-to-Matte’ step, the accuracy of the alpha mattes highly depends on the used trimap-based network, and I wonder whether it can meet the standard required for image matting annotation. As shown in Figure 1, the generated ‘GT’ alpha mattes do not seem to be good enough.
- Limitations of the natural scenarios and image resolution. The COCO-Matting mainly contains scenes with humans and does not adequately cover natural scenes relevant to matting, such as transparent or mesh materials, or any semantic category. It is OK to focus on human matting, but the dataset contribution seems overclaimed, so as the title of the paper. This work mainly deals with human-specific natural scenes but not all real scenarios in the wild. Another potential limitation is that matting generally requires high-resolution training data, but the COCO-Matting may not meet this prerequisite.

ii) Method
- The performance improvement comes at the cost of usability and flexibility. Compared with the MAM, Matting Anything, and SmartMat, the proposed method is specifically optimized for box prompts. It is not surprising to see the improvement under this specialized optimization. Despite the improvements in experiments, SEMat does not support other forms of prompts such as masks, points, and scribbles, reducing flexibility and usability in interactive image matting.
- Limited technical depth. The proposed enhancements of the SAM mainly involve off-the-shelf techniques such as LoRA and Trimap classification loss. These attempts are either generic or well-justified in the field. The fusion of the box prompt information at the input level, at the feature level, and at the output level is good, but not enough.

iii) Experiments
- Unfair comparison with SmartMat. Their backbones are different. How would the SEMat perform if replaced with DINOv2 as the backbone?
- The current experimental results may not support the SEMat as an effective, generic NATURAL image matting approach in the wild. The visualizations mainly focus on portraits and animals with similar hair/fur patterns. Aside from the cup shown in Figure 4, what is SEMat's performance on other natural images?
- It is not sufficiently convincing to compare the SEMat with the InstMatt under the current setting. The SEMat requires additional Bbox input from Grounding DINO for instance matting; some improvements may be attributed to the good localization performance. Also, conducting the ablation study on the HIM2k may not be a good choice. The ablation study should focus on justifying the matting ability of the model.
- Interactive matting with Bbox prompts has already been integrated into the latest version of Adobe Photoshop. Could it be compared with the proposed method?

iv) Minor issues
- There is a typo error in Line349, where ‘Specially, We…’ should be ‘Specially, we…’

**Questions:**

- Overall, the contributions of the work are not universal and may be merely interested to the image matting community. Compared with ICLR, the paper might be better suited for a computer vision conference.
- It is recommended to reduce the title to natural HUMAN matting.
- Manual post-screening to correct any potential annotated errors is encouraged for the ‘Accessory Fusion’ step.
- To showcase the reasonableness of low-res COCO-Matting on high-res alpha matting, it would be good to add an experiment to validate the model trained on COCO-Matting but testing on images of varied resolution, particularly on high-res natural images.

**Details Of Ethics Concerns:**

N.A.

---

### Note · Authors · 2024-11-15

I have read and agree with the venue's withdrawal policy on behalf of myself and my co-authors.